# Influence of Sit-Stand Tables in Classrooms on Children’s Sedentary Behavior and Teacher’s Acceptance and Feasibility: A Mixed-Methods Study

**DOI:** 10.3390/ijerph19116727

**Published:** 2022-05-31

**Authors:** Paula Schwenke, Michaela Coenen

**Affiliations:** 1Institute for Medical Information Processing, Biometry, and Epidemiology—IBE, Chair of Public Health and Health Services Research, LMU Munich, Elisabeth-Winterhalter-Weg 6, 81377 Munich, Germany; coenen@ibe.med.uni-muenchen.de; 2Pettenkofer School of Public Health, Elisabeth-Winterhalter-Weg 6, 81377 Munich, Germany

**Keywords:** sit-stand table, primary school, secondary school, physical activity, sedentary behaviour, school-based intervention

## Abstract

Children spend over 70% of their school day sitting, most of the time in the classroom. Even when meeting physical activity guidelines but sitting for long uninterrupted periods, children are at risk of poorer health outcomes. With an approach to create an active learning environment through the implementation of sit-stand tables, this exploratory mixed-methods study aims to evaluate a holistic concept for reducing sedentary time in schools by implementing sit-stand tables as well as to examine the feasibility and didactic usability in classroom settings. Children from eight German schools aged 7 to 10 in primary schools and 11 to 13 in secondary schools (*n* = 211), allocated into control and intervention groups, were included in the study, as well as teachers (*n* = 13). An accelerometer was used as a quantitative measure to assess sitting and standing times and sport motoric tests were taken. Qualitative interviews were performed with teachers regarding feasibility and acceptance of the sit-stand tables. Independent *t*-test analysis adjusted for age, sex and school type found that sitting times of children in the intervention group could be reduced (by 30.54 min per school day of 6 h, *p* < 0.001) within all school and age levels. Overall, implementing sit-stand tables in classrooms serves as a feasible and effective opportunity to reduce sedentary behaviour and create an active learning environment.

## 1. Introduction

A sedentary lifestyle and a lack of physical activity among children increase the risk of negative health effects in childhood and negative health-related consequences in further life [1,2]. Studies have shown that preventive strategies for adult inactivity should start through the establishment of movement patterns during childhood [3,4,5]. Lack of exercise increases the risk of metabolic diseases such as obesity, metabolic syndrome, diabetes mellitus [4,6], posture problems [7], back pain [8] and decreased academic achievement [9,10,11]. Guthold et al. conducted an international pooled analysis with 1.6 million adolescents and pointed out that in Germany, more than 80% of the participants do not reach the specified WHO activity guidelines [12]. Instead, children already spend over 70% of their school day sitting, most of the time in the classroom [13,14]. Even when meeting physical activity guidelines but sitting for long uninterrupted periods, children are at risk of poorer health outcomes like an increased risk of cardio-metabolic disease as well as a greater risk of higher waist circumference and BMI [9,15,16,17,18,19].

Schools are an important setting for health promotion activities. They offer the opportunity to counter inactivity and increased sitting behaviour at an early stage as well as enabling children of different socioeconomic backgrounds to become more active in their everyday lives while targeting health inequalities at the same time [20]. Thus, it is known that it is possible to include feasible strategies in daily school life to reduce and break up inactive times using different classroom-based interventions like active breaks, didactic methods and adapted classroom environments [21]. With sit-stand tables and approaches to dynamic learning, there are holistic strategies to increase and strengthen children’s health literacy at a young age. The evidence indicates that sit-stand tables are an effective intervention to increase energy and caloric expenditure as well as to relieve stress on spinal structures that may build up with use of traditional desks [4,22,23]. Furthermore, increasing standing times during school hours have been hypothesized to have a positive impact on cognitive and mental functions and overall wellbeing [24,25,26].

To date, the evidence of associations between objectively assessed sitting and standing times regarding physical fitness and cognitive outcomes based on the implementation of sit-stand tables is limited to small numbers of participants in the previous studies (range *n* = 5 to 31, [27,28]). Moreover, evidence of the effect of sit-stand tables implemented in school settings and related didactic concepts is mostly restricted to Australia, USA, the UK and New Zealand [27]. Due to the different school systems, a transferability of these studies to schools in other countries such as Germany is unclear. Most studies to date also focused on primary schools and few studies have examined the effects of sit-stand tables in secondary schools [29]. Furthermore, the existing studies only present limited data on teachers’ experiences regarding the feasibility and acceptance of implementing sit-stand tables in classrooms.

Therefore, the aim of this study is to evaluate a holistic concept for reducing sedentary time in schools by implementing sit-stand tables as well as to examine the feasibility and didactic usability in classroom settings. Specific aims are to (1) analyse sitting and standing times in classrooms with and without sit-stand tables, (2) explore which children in classrooms with sit-stand tables are more likely to change between sitting and standing times during lessons with a special focus on sex, BMI, sport activities (outside the school setting) and fitness level, and (3) explore intervention-group teachers’ views on the feasibility and acceptance of the intervention.

## 2. Materials and Methods

### 2.1. Study Design

This school-based, two-arm, controlled, mixed-methods study was carried out between August 2020 and July 2021 in Munich, the capital city of Bavaria, as well as the metropolitan area of Munich, Germany. Due to the COVID-19 pandemic, there was no data collection between 14th of December 2020 and 1st of March 2021, as well as between 19th of April and 21st of May 2021, because of school closures and home schooling in Bavaria.

Classes were allocated to intervention and control groups where children of intervention groups used sit-stand tables and children of control groups relied on their traditional working tables. Quantitative data included measurements of sitting and standing times as well as sport-motoric tests. Due to limited accelerometer stock, sitting and standing data was accomplished within a nested study. Qualitative assessments included interviews with teachers from intervention groups.

The study was carried out in accordance with the Declaration of Helsinki; furthermore, the study protocol was approved by the ethical commission from the medical faculty of the LMU Munich (20-0785, 16 October 2020).

### 2.2. Intervention

The sit-stand tables (Rocket-Table, Figure 1) implemented in the intervention groups are height-adjustable between 53–83 cm for 1st–4th grade (RT3000) and 65–106 cm for 4th–8th grade (RT6000) with a mechanical lever and steps of 19 mm to match the individual’s ergonomic working position. After the implementation of the tables, the children were shown how to work with the height-adjustment and then were able to choose sit or stand positions freely during the lessons without the need of support. No targeted daily sitting or standing aims were prescribed to teachers or children, and the height of sitting and standing positions was not assessed. The children kept their traditional chair for sitting times.

### 2.3. Study Sample and Participants

Children from primary schools aged between 7 and 10 years and children from secondary schools aged 11 to 13 years were included in this study. Recruitment was conducted class-wise in eight schools: four in Munich and four in the metropolitan area of Munich, Germany. All schools were randomly selected based on the first author’s personal contacts. Thus, classes were recruited when the head of schools and teachers of the respective classes were likely to participate; no exclusion criteria were used.

As presented in Table 1, the number of participating classes ranged from one to three per school. Three schools had each one intervention and one control class, the other schools had mixed classes with children who used sit-stand tables and children who worked at traditional working tables within one class. In total, five schools were part of the nested study (see also Figure 2).

In addition, teachers of the intervention and mixed classes were included in the qualitative assessments; no exclusion criteria were used. More than one teacher per intervention group was interviewed in two secondary schools, as the classes had more teacher changes in contrast to primary schools with one head teacher per group. All included teachers were asked to sign an informed consent prior to the interview.

As presented in Figure 2, parents of 255 children were informed during parents’ evening sessions and by provision of printed study information and were asked to sign an informed consent. Additionally, the children were informed prior to the first intervention and were asked to sign an informed consent as well. Due to dropouts regarding school changes, missing informed consent or being absent at baseline testing, in total, 211 children were included in the study at baseline. Consequently, 144 children were included within the nested study; thereof, 79 children were allocated to the intervention group and 65 to the control group. Due to COVID-19 restrictions in Bavaria regarding school closures and home schooling, not all children participated at all three data collection time points.

## 3. Measurements and Data Collection

### 3.1. Quantitative Data

The baseline measurements involved an assessment of anthropometric data including weight, height and age, which was taken with a digital scale and a measuring tape. The BMI was calculated (kg/m^2^) and classified in accordance with age- and sex-adjusted BMI percentiles [30]. Furthermore, the fitness level was measured with sport motoric tests with one child at a time by two researchers within the school premises. To ensure the reliable and systematic performance of the tests, instruction guidelines were taken from the standardized Deutscher Sportmotoriktest (DMT Karlsruhe, Germany [31]; German sport motoric test). Physical performance measures included: (1) long jump with closed feet; (2) single leg stand; (3) forward bend with straight legs were measured twice to predict temporary variations. The forward bend was measured while the children were standing on a bench. All data were processed based on the calculated means of both measurements and categorised within an age- and sex-adjusted standardised scoring system into low (<30), normal (30–70) and high (>70) fitness level [31]. Furthermore, children were asked about their sport activities outside of school.

The data on sport activities (outside the school setting) were assessed during the measurement of the anthropometric data with a short questionnaire.

Sitting and standing times were assessed within a nested study due to limited accelerometer stock. Overall, accelerometer wear-time data was conducted with 163 children, who were allocated into intervention and control groups and completed at least one and up to three accelerometer assessments (T0, T1, T2). The assessments for each time point (T0, T1 and T2) were not conducted in parallel in all schools because of organizational reasons. On each time point, students wore the accelerometer for 5 days; the time interval between T0, T1 and T2 per school was mostly 4 weeks, and sometimes slightly longer due to the COVID-19-related lockdowns. The sample size difference between assessment points is due to the fact that home schooling during the COVID-19 pandemic made children’s attendance in class unpredictable. Data within both groups were conducted by using a Move 3 accelerometer (Movisens, Karlsruhe, Germany, [32]) worn on the right thigh for five consecutive days. The accelerometers used have demonstrated reliability and validity for use in setting-related sit and stand studies [26]. Teachers and children were instructed by the researchers about the exact wearing position and handling of the accelerometers. The exact wearing position of the accelerometer was on the highest position on the right thigh. The accelerometer was fastened to an adjustable and five-centimeter-wide elastic band which was worn over the trousers. Only the times spent in the classroom were included in the analyses. During outdoor breaks, the accelerometers were left inside the classroom. All the accelerometer data were downloaded and analysed by using manufacturer proprietary software [33]; the data processing was managed with Excel. The Move 3 accelerometer collected data in 10-s epochs; non-wear time and time spent outside the classroom could be defined based on the provided timetables and information from the teachers. Provided non-wear times were additionally checked with the Movisens data viewer software [34]. To prepare comparable data, wear-time differences between the schools and groups were eradicated by summing up the collected data for each participant and dividing it by the number of lessons in which valid data was obtained and then calculating the means per school hour (45 min).

### 3.2. Qualitative Data

Following the last assessments (T2), teachers of intervention and mixed classes were interviewed to report on their experience about the feasibility and acceptance of sit-stand tables in the classroom setting. Qualitative data collection was conducted by one researcher, using a semi-structured interview guideline addressing the four topics (feasibility, usability, impact and integration into classrooms). Interviews were performed online with the conference software Zoom and recorded with a voice recorder, verbatim transcribed into a Word document and anonymized. Within the interviews, the teachers were asked about observed changes in class dynamics since the sit-stand tables were implemented. Furthermore, they were asked about their perceptions of the impact of standing positions during lessons on children’s behaviour and concentration ability as well as other positive and negative aspects and usability and feasibility observations during a regular school day.

## 4. Statistical Analyses

### 4.1. Quantitative Data

Sensitivity analysis and sample size calculations were conducted with the G*Power software [35]. A power (1-ß err prob) of 80% was assumed, as well as a significance value of 0.05α (err prob) or lower. In order to achieve a high effect, the sample size calculation estimated a total number of at least 128 participants.

Descriptive analysis was performed for all variables applying absolute and relative frequencies and mean and standard deviation, respectively. For comparison of children in intervention and control groups at baseline, chi-square statistics and independent *t*-test analyses were performed.

Independent *t*-tests were also performed to analyse total sitting and standing times clustered in intervention and control groups as well as stratified for each school and merged in primary and secondary school categories. Average sitting and standing times per class were scaled to 45 min school hours to allow for comparability. For further independent *t*-test analyses, sitting and standing times in intervention and control groups were clustered by sex, BMI percentiles, sports and fitness level.

The significance level for all tests was marked in bold below 0.05. Furthermore, the variance homogeneity was checked using Levene’s test.

All data was analysed using the statistics software SPSS (IBM SPSS Statistics 27.0.0 [36], IBM, Armonk, NY, USA).

### 4.2. Qualitative Data

Qualitative data was analysed following the content analyses according to Mayring [37,38] and coded using the software MAXQDA by the first author (PS). Open coding was performed by a second researcher (LS), rather than the one who conducted the interviews, to ensure the rigor of the analyses. As a second step, the first author completed analyses by using selective coding to summarize the overarching themes and sub codes as well as adding frequency counts for pattern recognition in responses.

After analysing the collected data, schools were provided with a result summary as well as with additional material to implement dynamic and active teaching by using sit-stand tables.

## 5. Results

### 5.1. Sample Characteristics

Descriptive statistics for the study sample are shown in Table 2. In total, 211 children, 42.2% female (n = 89) and 57.8% male (n = 122) with a mean age of 9.06 (SD = 1.7), ranging from 7 to 13 years, were involved at baseline (T0). One hundred thirty-nine children (65.9%) were assessed within primary school settings, whereas seventy-two children (34.1%) were included from secondary schools. In total, 60.7% of the observed children (n = 128) were inside the normal weight percentiles. Thirty-three children (15.6%) were overweight, of which thirteen children (13.3%) were in the intervention and twenty children (17.7%) in the control group. Twenty-seven children had obesity (12.8%), of which 17.3% were in the intervention and 8.8% were in the control group.

Furthermore, 135 (64.0%) of the children reported participating in sport activities outside of school (n = 51 (52.0%) within the intervention group; n = 84 (74.3%) within control group). In total, 14.3% of children (n = 14) with a low fitness level participated in the intervention group, compared to 1.8% (n = 2) in the control group. One hundred seventy-three children (82%) showed a normal fitness level and twenty-two (10.4%) showed a high fitness level.

Regarding the described characteristics, there were no significant differences between children in the intervention and control groups with the exception of fitness level. Independent *t*-tests revealed that there were more children in the control group with a higher fitness level compared to the intervention group (*p* < 0.003).

### 5.2. Sitting and Standing Time

Table 3 summarizes the comparison of sitting and standing times of the intervention and control groups, stratified for schools and primary and secondary school setting. Accelerometer-measured sitting and standing data was different between intervention and control groups within all schools (*p* < 0.001). Overall, participants sat on average 36.04 min (3.96) and stood 9.34 min (4.01) per 45 min in the intervention group and 41.2 min (2.21) and 4.11 min (2.22) in the control group.

The stratified analysis regarding primary and secondary schools showed that children in primary school settings sat 35.82 min (3.89) per 45 min in the intervention group and 41.04 min (2.21) in the control group. Children in secondary schools sat 35.26 min (1.60) in the intervention group and 44.64 min (0.71) in the control group. Therefore, both school settings had similar sitting and standing times in the intervention groups but differed in the control groups.

Nevertheless, there was still an observed variability among individual children within the groups (see Appendix A). The average sitting and standing times calculated for 45 min school hours ranged from 32.41 min sitting and 12.97 min standing up to 45 min sitting and 0 min standing in the control groups and 13.27 min sitting and 32.39 min standing compared to 43.24 min sitting and 2.19 min standing in the intervention groups. This also shows that the variety within individual children in the intervention groups was higher (sitting 29.97 min difference; standing 30.2 min) than in the control groups (sitting 12.99 min; standing 12.97 min).

### 5.3. Association of Sedentary Behaviour Regarding Sex, BMI, Sport Activities (Outside the School Setting) and Fitness Level

The results of sitting and standing times in relation to sex, BMI and sport activities (outside the school setting) (Table 4) show that there was no difference regarding the mentioned characteristics. Only the fitness level variable revealed a significance difference (*p* < 0.001) in sitting and standing times for both groups within children who had a normal (IG: 35.81 (3.72), CG: 41.25 min (2.18)) and a higher fitness level (IG: 32.36 (4.8), CG: 39.94 (1.86)) compared to children with a lower fitness level (IG: 39.49 (2.01), CG: 44.52).

### 5.4. Results from the Teachers’ Perspective in Intervention Group

The semi-structured interviews with the teachers of the intervention group revealed nine overarching themes (allotted to four topics), which were further divided into sub codes as shown in Figure 3.

In total, 13 teachers from the intervention group participated in the interviews. They were mostly female (76.92%), with four being <35 years, five being 36–45 years and four being >45 years. Of these, 61.54% were teaching in secondary classes and the span of hours per week while teaching in intervention groups ranged from 3 to 26 (mean 14.54, median 16). Ten of the interviewed persons were teaching main subjects (Maths, German) and three were teaching sports.

The answers given by the teachers are presented in the following structure regarding the four topics shown in the coding scheme (Figure 3).

### 5.5. Impact from the Teachers’ Perspective

After implementing sit-stand tables in the intervention groups, teachers reported an improvement in the teaching atmosphere due to more dynamic lessons and motivated children: “Lessons are more dynamic and children are more motivated because of changing positions, they also motivate each other” (I4, L5; I3, L72). Thus, children who had difficulties sitting for long periods could now be integrated into the class more easily. Teachers could tell them to stand up instead of sending them outside the classroom or being disrupted more frequently: “There was no more fidgeting, I could tell those pupils to stand up” (I1, L38–40); “There were fewer children, I had to send out to release energy” (I4, L11). Another point mentioned was teaching at eye level, which was experienced as helpful in fostering a better child–teacher connection: “From the teacher’s point of view, it was very pleasant to be opposite standing persons, at eye level, good for the pupil-teacher relationship” (I3, L39). Regarding the practicability in class, most teachers reported that it worked very well, being especially beneficial during the Covid-19 pandemic as children had to stay at their single tables to keep the required distance but could still change positions regularly: “Sit-stand tables were very practical during the Covid-19-pandemic, because children need a lot of movement and I usually include many movement phases and now they have to keep their distance all the time and I can’t do any more movement games, but now they can stand up” (I6, L13). Jealousy and the need for setting up a rotation structure only appeared within classes which were not fully equipped with sit-stand tables.

The impact on the children regarding health-related factors was reported to be positive among all interviewed teachers. The teachers mentioned an improved body awareness: “It is good to be able to decide for yourself, this rises the health literacy” (I7, L5); “regular change between positions strengthens the muscles” (I5, l23). Specifically reported were positive outcomes regarding “restless children, as energy can be channelled differently when standing up.” (I9, L6) because “any movement helps children with attention problems.” (I6, l46).

### 5.6. Usability from the Teachers’ Perspective

The analyses of the usability of implementing sit-stand tables in classrooms were conducted by asking about the frequency of use, the handling of height-adjustment as well as the quality of the tables. The observed use rate ranged from “twice every 45 min” (I5, L36–37) to “some children did not move the tables at all without being asked to do so.” (I1, L60) and the “Motivation was very high at the beginning, then it dropped a bit, not so easy due to Corona because they were often at home” (I2, L9). However, most teachers reported that there were children who stood up regularly while some children only when teachers stimulated them to do so. A typical characteristic of those who regularly used the possibility to stand were reported inconsistency. “The type of users is very mixed, both children who were previously quiet and those who are very fidgety use it.” (I10, L6) and “It was not possible to predict who would use it more often. So not only those who generally move more or are more active.” (I6, L20). Contrary to the observation of inconsistency, some teachers reported that “more active children asked more often if they could work standing up.” (I4, L17) and “It was more likely to activate those who […] were physically fit” (I4, L15) as well as “children who had weak motor skills and body perception, I had the feeling that they did not like to stand up so much.” (I6, L22). In addition, two teachers mentioned that “boys used the opportunity to stand up more often than girls” (I2, L11; I10, L6).

### 5.7. Feasibility from the Teachers’ Perspective

Therefore, most interviewed teachers (*n* = 10) had a guided structure of use for the first two weeks. Within this timeframe, children were allowed to stand up during structured “stand-up times” or during quiet or partner work. After two weeks, children were allowed to decide for themselves when to sit or stand and some teachers left it free to decide from the beginning: “I didn’t want to prescribe it, each child is individual, maybe a child wants to get up after 5 min already.” (I6, L12). Regarding the handling of the height adjustment, teachers were “surprised how quickly the children did it without any problems on their own” (I6, Z13) and “over time, all the children, even those with weaker motor skills and physical abilities, were able to do it.” (I6, L29) while it was “not too loud” (I7, L15). Other aspects included it being great to have height-adjustable tables when working with different age groups and some teachers used a table for themselves. One teacher also reported that “Some children found it difficult to think and feel when it might be good to stand up” (I7, L27) and that “Sometimes the table-top was still too high when sitting down again, so teacher had to help with the right height” (I6, L11). Regarding the design of the sit-stand tables, all teachers mentioned similar aspects, such as “Stable and not too heavy” (I9, L9), “we have often used the tablet rail in the front of the table top, it would be cool if you could also put books in there.” (I6, L41) and “very stable design, nothing rattles when they are adjusting the height.” (I1, L28). Answers given by the teachers regarding negative or disturbing factors while using sit-stand tables in classes were that one third of the teachers feared that children would play around with the adjustment or cause disruptions during lessons. All teachers mentioned that this fear quickly subsided; there were few individuals who changed the position very often, disturbing others. Additionally, teachers reported visibility problems if it was not the case that everyone had a movable and height-adjustable table, but also that this could be managed through changed seating orders or with the movability of the table. Further ideas to improve learning with the sit-stand table were “Hooks on the table for backpacks would be good” (I12, L6), “a drawer instead of bookshelf under the table-top would be nice” (I8, L6) and “It would be good if the table-top was a bit larger, otherwise things sometimes fall down more easily.” (I8, L6).

### 5.8. Integration in Classroom Setting

To integrate sit-stand tables in everyday lessons, teachers were asked how best they could be combined with didactic methods. Answers were structured into three groups of didactic interaction: 1. silent work; 2. group work; and 3. Class set-ups. In silent work phases, teachers mentioned that children “could quickly turn around with the table facing the wall and were not distracted by others” (I2, L42) and “Children could quickly move around or push the table out into the hallway to work silently” (I12, L18). Those cited both refer more to the easy movability rather than the height adjustment. However, as already pointed out regarding experienced effects, “children were motivated to choose position often during silent work times” (I4, L50; I11, L9). Regarding group works, teachers reported the movability to be very practical, because options like a standing corner for specific assignments or a set-up with different stations was possible; additionally, the “Tables can be easily moved to change the seating order, e.g., in a circle, nice that table-top corners are rounded so they can be arranged in semicircles.” (I6, L62). Some teachers also used the tables as presentation desks in front of the class, which was “helpful for children who are not so confident, because it gives them security.” (I9, L9). “Even if you only have a few desks, you can put them in the back and everyone who can’t sit any more was allowed to work there.” (I4, L64). Finally, teachers were asked about additional movement ideas for future wholistic movement concepts. In-class concepts included active games such as jumping in maths or working with clapping, e.g., in a German lesson to better remember difficult words. Further ideas included running dictations, reading walks or exercises which could be performed behind the school table. Out-of-class concepts contained lessons outside in nature or movement events such as running a targeted number of kilometres together.

## 6. Discussion

To the authors’ knowledge, the present study is the first in Germany to assess a large-scale, multi-school, multi-grade cohort of elementary and secondary school children regarding the influence of sit-stand tables on sitting and standing times as well as to examine which children are more likely to use the provided sit-stand tables regarding sex, BMI, sport activities (outside the school setting) and fitness level and to explore intervention-group teachers’ feasibility and acceptance.

Overall, our findings show that sitting and standing data were significantly different between intervention and control groups and sitting times could be reduced within all intervention groups. Summing up the presented sitting and standing times for an average school day of 6 h, it is shown that sitting time within intervention groups could be reduced by 30 min and 54 s per day, while standing times could be increased by 31 min and 23 s. Similar findings have also been reported in studies already conducted in other countries, as there were significant reductions in sitting times: between 9.4% or 44 min/day [39], 25% [13], 26 min/school day [40] and 64 min/day [27]. The comparability of these data across the studies is unclear due to the differing amounts of hours spent in school per day. Therefore, we calculated sitting and standing times per measured school hour (45 min) so that a comparability between participating schools in this study could be achieved. However, comparisons with existing research results is difficult since other school examinations have all been performed in other countries than Germany, with different school systems and few studies, to our knowledge, have examined classrooms which were fully equipped with sit-stand tables [4,13,28,39,41,42].

The results are discussed in relation to the analysed characteristics as follows.

In average, all children in the intervention group used the possibility to change between sitting and standing equally despite their sex contribution, BMI status or whether they participate in sports outside of school. We showed that within this study, children with higher fitness levels used the possibility to change between sitting and standing more often, whereas children with a lower fitness level used the sit-stand tables less often. However, more research is needed that includes other activity parameters inside and outside of the school setting to show whether sit-stand tables are used more often by children who are physically fit compared to less active ones. Therefore, specific behavioural interventions might be helpful to further target an increased use of sit-stand tables among children who have a lower fitness level. Additionally, an observed variability in sitting and standing times among individual children might be targeted with additional behavioural interventions. Further investigation could focus on more varied factors to identify the characteristics of children who were more sedentary despite having the possibility to stand.

To our knowledge, this study is the first to show that children, regardless of their BMI status, use sit-stand tables, which might be a promising first step towards a strategy to reduce obesity among children. Contrarily to this study, other research has shown that overweight children in sit-stand table classrooms do not necessarily move at the same rate as their normal-weight peers [4]. However, generally, research showed a significant increase in caloric expenditure after implementing sit-stand tables [40], which can be a promising basis for an intervention benefiting all children more equally [4].

In addition to the potential effect on BMI status, further studies have reported positive effects when reducing or interrupting sitting time in childhood. Sedentary behaviour and uninterrupted sitting for more than 15–20 min have been associated with cardio-metabolic outcomes in adulthood [19] and sitting for 5–10 min has been negatively correlated with inflammatory markers in children [43]. In addition, research has shown that reducing sedentary behaviour improves fitness levels, academic performance and overall cognitive development [9,17,43]. Due to recurring COVID-19-related school closures, the initially planned measurements regarding concentration ability could not be carried out as planned. However, other studies have shown that there are correlations related to cognitive performance among children using sit-stand tables. Colley et al. found that the utilization of sit-stand tables was largely associated with improved working memory capabilities. In total, there was an improvement of 7–14% in cognitive performance across several executive function and working memory tasks [44]. Furthermore, research showed that environmental changes in classrooms increase children’s cognitive functions, which drive their cognitive development and impact educational outcomes [44], and that children who make more steps have quicker inhibition response times [45]. However, in contrast to the improved cognitive performance, research also revealed that more sitting time correlates with higher lapses of attention [45] or better sustained attention [46]. Additionally, children were less accurate in their responses while standing than sitting [45]. Despite the lack of data on concentration in our study, our qualitative data analyses in teachers’ interviews revealed a perceived improvement in concentration ability as well as prolonged concentration phases.

The analysed feasibility and acceptance of integrating sit-stand tables is consistent with previous studies, which had found that most students and teachers in primary [41,42,47,48] and secondary school settings [29] were pleased about the opportunity to learn more dynamically. In this study, and also in previous studies, teachers reported less disruptions after implementing sit-stand tables in classes, because students behaved better in a dynamic environment while being happier and more motivated [47]. Another study showed that sit-stand tables limited occurrences of misbehaviour, such as bothering other individuals or making distracting movements or noises [49].

A novel aspect of our study was that teachers reported an improved body awareness and that teaching at eye level was experienced as helpful for an improved child–teacher-connection. Especially during the COVID-19 pandemic, single sit-stand tables were experienced as very helpful in keeping the statutory distance and still could allow children to move when changing between sitting and standing. Children of all observed age categories could manage the height adjustment after some introduction time without problems, which is an important prerequisite for the increased active behaviour.

Few adverse impacts were reported, such as teachers’ fear of disruptions by using the sit-stand tables in the class room. However, teachers reported that these fears were not confirmed during the intervention. Furthermore, visibility problems were reported, which could be solved through changing the seating order or moving tables, if needed, to the right or left. This is consistent with studies already conducted as they also mentioned a rearrangement of the classroom as a solution to visibility problems [50].

Additionally, interviewed teachers had a wide range of ideas about integrating sit-stand tables into didactic methods, such as partner or group works as well as silent work phases or presentation desk options. However, there are few studies, to our knowledge, observing relations between sit-stand tables and didactic options with concrete results [51]. Other research has shown that teachers play a crucial role in motivating children in the long term [50] and when placing only one sit-stand table in a classroom, children stay motivated for a short time only [52]. Therefore, the collected ideas of active teaching styles together with sit-stand tables will be helpful in advancing the holistic and sustainable effectiveness of the implemented intervention. Accordingly, teachers’ participation in the study and their brainstorming about didactic integrity might enable them to train others interested in using sit-stand tables. After this study was completed, summaries of the results were presented to schools, teachers and funders to make dynamic and active teaching more likely. Therefore, schools were provided with short and active videos for school breaks as well as with different exercises and movement games to perform inside the classroom. Additional material regarding school-specific results, ideas and didactic possibilities which resulted from the teacher interviews were specially prepared for each school to use for future holistic movement concepts.

Previous studies also show that sitting and standing times might differ in relation to morning or afternoon lessons [13]. In our study, we could not separate mornings and afternoons due to inconsistent school structures and alternating lessons caused by the COVID-19 pandemic. For future research, it will be interesting to observe how sedentary behaviour relates to the time of day. Another important aspect which should be considered for future research is how many school hours per week children were able to learn in a room with sit-stand tables. Especially in the secondary school setting, students more often change rooms. For example, Sudholz et al. mentioned that in their study, students only had one to two lessons in the classroom with the sit-stand tables per week [13], which appears not to be comparable with other studies. Furthermore, studies could examine the associations between different didactic concepts, the amount of sport offered in school and sedentary behaviour.

### Limitations and Strengths

The major limitations of the present study are: (a) the inconsistent data material with missing values due to COVID-19 interruptions; (b) drop-out rates; and (c) the non-randomized study design. (a) The COVID-19 pandemic with closing of schools or established part-class lessons made the planned timing of T0, T1 and T2 very difficult. Additionally, the planned concentration tests (KKA: Kaseler-Konzentrationsaufgabe for primary school, D2R for secondary school, [53,54]) could not be carried out completely and were therefore excluded from the main data analyses because of a high number of dropouts, missing data and COVID-19 related unstable learning and school conditions. Therefore, independent *t*-tests were conducted within this study rather than a multivariate analysis. (b) The COVID-19 pandemic was most impactful in causing continual dropouts which reduced the total sample size. (c) Due to the dependence of schools and teachers’ willingness to participate, the study could not be conducted in a randomized manner. Even though the participating schools were in different areas, the demographics of the included children may not be representative of other school settings. More research is needed to understand how different demographic and school characteristics would influence the effectiveness of sit-stand tables.

Despite these limitations, the strengths of the present study were the inclusion of different schools, grades and school settings and the mixed-methods design. Thus, it could be confirmed that a reduction in sitting time was possible within all eight participating schools. Even though measurements in secondary schools were more challenging due to increased changing between different classrooms, all the data could be compared after calculating per school hour (45 min). Another key strength is the high number of participants and the possibility to fully equip some classrooms.

## 7. Conclusions

In summary, the presented results extend the current knowledge regarding the effects and feasibility of sit-stand table interventions to increase standing and reduce sitting times during school lessons in German primary and secondary schools. For all children, regardless of their sex, BMI status or sport activity, reduced sitting times were reported by implementing sit-stand tables; this intervention may also be a successful option to target health inequalities. As the study showed a good feasibility and didactic integrity related to using sit-stand tables within the classroom setting, whilst considering the health risks associated with physical inactivity, schools and stakeholders may rethink traditional approaches to teaching.

## Figures and Tables

**Figure 1 ijerph-19-06727-f001:**
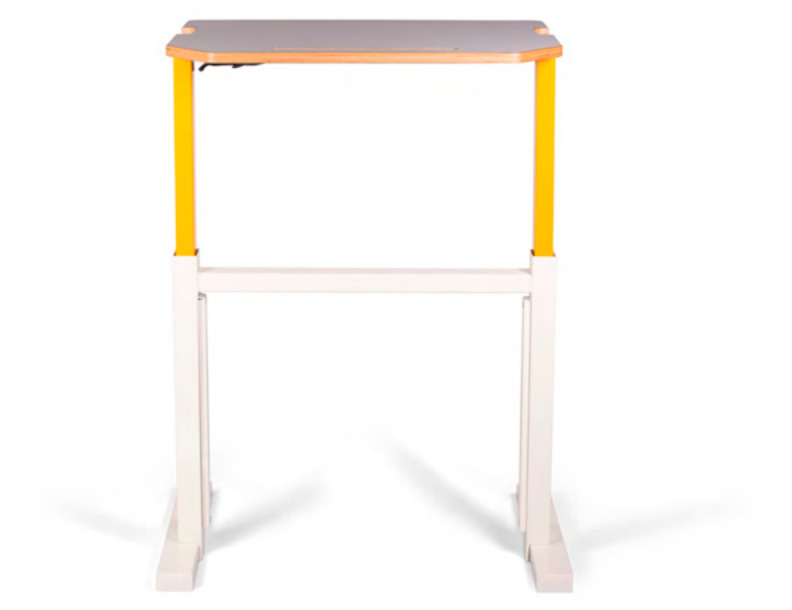
Model of the used sit-stand table created by Rocket-Table GmbH.

**Figure 2 ijerph-19-06727-f002:**
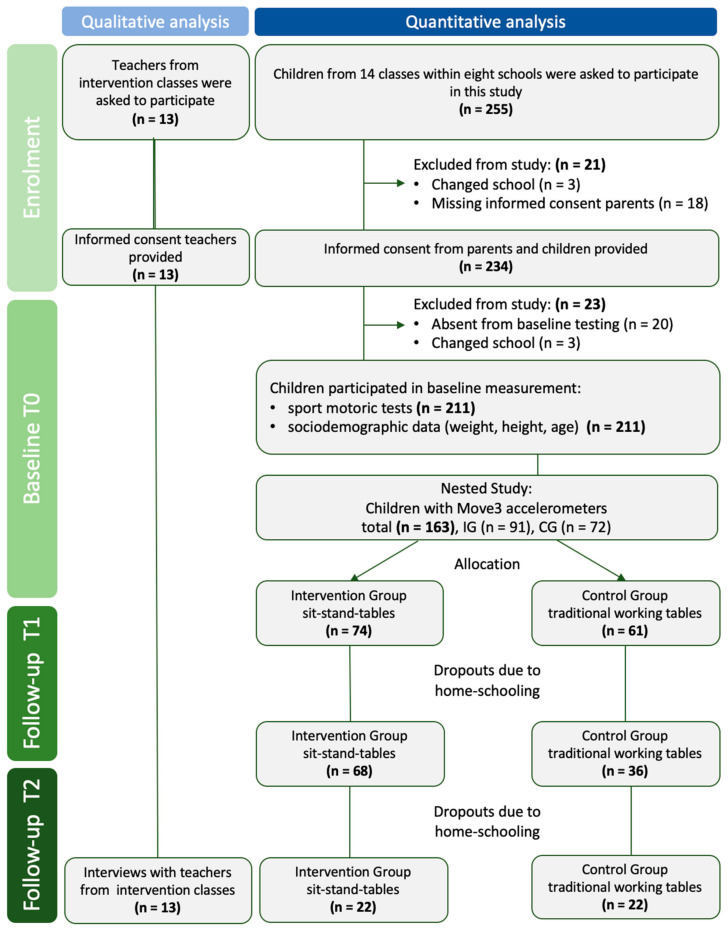
Study participants flowchart.

**Figure 3 ijerph-19-06727-f003:**
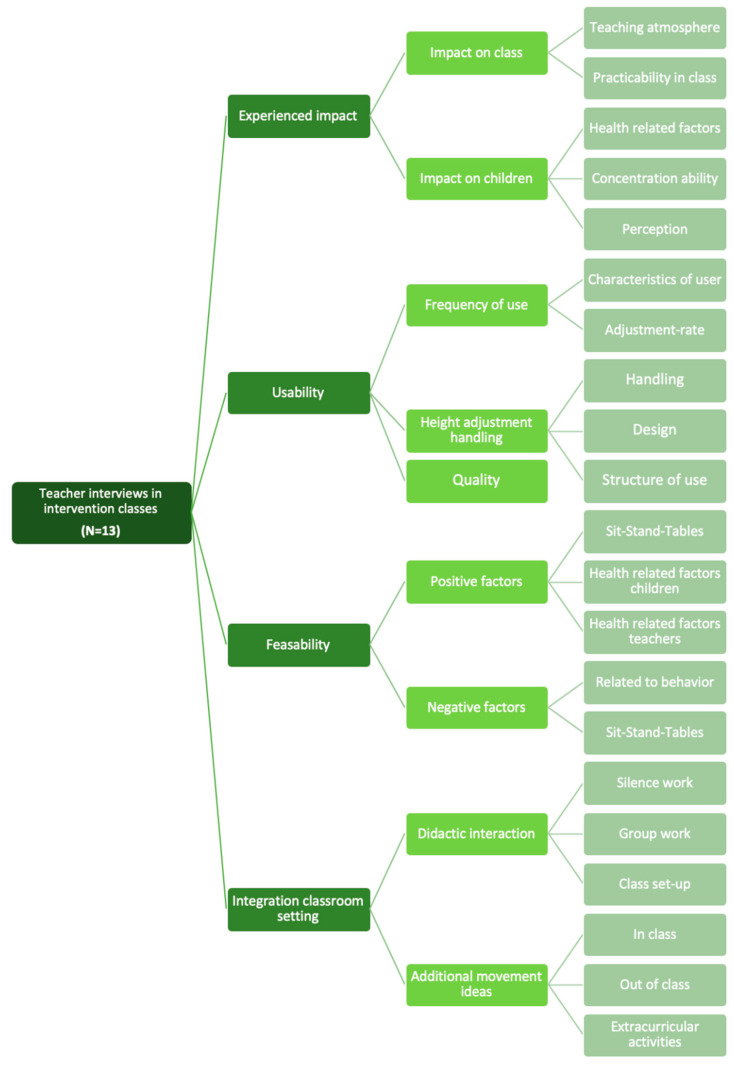
Coding scheme of the semi-structured interviews with teachers of the intervention group.

**Table 1 ijerph-19-06727-t001:** School characteristics.

School	Number of Participating Children Per School (*n*)	Primary School (P) Secondary School (S)	Location: Munich (M) Metropolitan Area (MA)	Number of Classes Participating in Study (*n*)	Schools Included in Nested Study (P)	Allocation to IG and CG *: Mixed Classes (M) Separated Classes (S)	Number of Interviewed Teachers from Intervention Classes (*n*)
1	31	P	M	2	P	S	1
2	34	P	M	2	P	S	1
3	29	P	M	2	P	S	1
4	34	P, S	M	3	P	M	3
5	16	S	MA	1	P	M	1
6	14	S	MA	1	-	M	2
7	32	S	MA	2	-	M	3
8	21	P	MA	1	-	M	1
Total	211	5 primary classes (*n* = 139 children) 4 secondary classes (*n* = 72 children)	4 schools in Munich (*n* = 128 children) 4 schools in metropolitan area of Munich (*n* = 83 children)	14	5 schools taking part in nested study	3 schools with one intervention and one control class 5 schools with mixed classes **	13

***** IG: Intervention Group, CG: Control Group. ****** Within three schools, the intervention and control groups were separated by class. Regarding the other five schools, there were children in each class who used sit-stand tables and children who worked at traditional working tables.

**Table 2 ijerph-19-06727-t002:** Descriptive characteristics of overall and intervention and control group sample.

Characteristics	All		Intervention		Control	
	N (211)	%/Mean (SD)	*n* (98)	%/Mean (SD)	*n* (113)	%/Mean (SD)
**Sex**						
Female	89	42.2	40	40.8	49	43.4
Male	122	57.8	58	59.2	63	56.6
**Age**		9.06 (1.70)		8.85 (1.79)		9.24 (1.61)
7	39	18.5	25	25.5	14	12.4
8	69	32.7	32	32.7	37	32.7
9	21	10.0	11	11.2	10	8.8
10	34	16.1	9	9.2	25	22.1
11	26	12.3	9	9.2	17	15.0
12	15	7.1	8	8.2	7	6.2
13	7	3.3	4	4.1	3	2.7
**Schools**						
Primary School	139	65.9	80	81.6	59	52.2
Secondary School	72	34.1	18	18.4	54	47.8
**Grade**						
2	75	35.5	51	52.0	24	21.2
3	52	24.6	17	17.3	35	31.0
4	12	5.7	12	12.2	0	0
5	49	23.2	10	10.2	39	34.5
6	23	10.9	8	8.2	15	13.3
**BMI Percentiles**						
Extreme Underweight (<3)	5	2.4	0	0	5	4.4
Underweight (3–15)	18	8.5	10	10.2	8	7.1
Normal weight (15–85)	128	60.7	58	59.2	70	61.9
Overweight (85–97)	33	15.6	13	13.3	20	17.7
Obesity (>97)	27	12.8	17	17.3	10	8.8
**Sport activities (outside the school setting)**						
No	76	36.0	47	48.0	29	25.7
Yes	135	64.0	51	52.0	84	74.3
**Fitness Level ***						
Low (<30)	16	7.6	14	14.3	2	1.8
Normal (30–70)	173	82.0	76	77.6	97	85.8
High (>70)	22	10.4	8	8.2	14	12.4

Abbreviations: SD: standard deviation, BMI: Body Mass Index (weight [kg]/height [m^2^]), N: number of cases in the total sample, n: number of cases in the subsamples of intervention and control group, * significant (<0.05) difference between intervention and control group.

**Table 3 ijerph-19-06727-t003:** Average sitting and standing time per 45 min school hour clustered in intervention and control group stratified for schools and primary and secondary school setting.

School	Grade		Intervention Group		Control Group		*t*-test (a)	
			*n*	M (SD)	*n*	M (SD)	95%-CI	*p*-Value
1	3	Time sitting	17	35.75 (3.73)	14	42.30 (1.11)	4.56, 8.54	<0.001
		Time standing		9.61 (3.78)		2.91 (1.10)	−8.71, −4.68	<0.001
2	2	Time sitting	22	32.26 (3.68)	12	40.97 (3.26)	6.12, 11.30	<0.001
		Time standing		13.25 (3.69)		4.36 (3.23)	−11.47, −6.30	<0.001
3	3	Time sitting	17	35.65 (2.81)	12	40.15 (1.23)	2.91, 6.09	<0.001
		Time standing		9.64 (2.83)		5.25 (1.19)	−5.97, −2.80	<0.001
4	2	Time sitting	11	37.58 (2.26)	8	40.27 (1.61)	1.11, 4.48	0.003
		Time standing		7.70 (2.20)		5.17 (1.60)	−4.06, −0.83	0.006
	4	Time sitting	9	37.87 (1.51)	8	41.53 (1.68)	1.63, 5.16	0.001
		Time standing		7.55 (1.48)		3.91 (1.88)	−4.95, −1.38	0.002
	5	Time sitting	10	37.57 (2.53)	10	44.36 (0.95)	5.16, 8.24	<0.001
		Time standing		7.76 (2.58)		1.13 (0.92)	−8.11, −5.01	<0.001
5	6	Time sitting	5	32.94 (7.57)	8	44.91 (0.47)	2.57, 21.37	0.024
		Time standing		13.12 (6.93)		0.44 (0.48)	−21.28, −4.10	0.015
Primary Schools	2, 3, 4	Time sitting	76	35.82 (3.89)	54	41.04 (2.21)	4.54, 6.81	<0.001
		Time standing		9.55 (3.95)		4.32 (2.22)	−6.90, −4.60	<0.001
Secondary Schools	5, 6	Time sitting	15	35.26 (1.60)	18	44.64 (0.71)	7.12, 10.30	<0.001
		Time standing		10.44 (1.60)		0.79 (0.70)	−10.74; −7.42	<0.001
Total	2–6	Time sitting	91	35.54 (3.96)	72	42.84 (2.21)	6.05, 8.26	<0.001
		Time standing		9.99 (4.01)		2.55 (2.22)	−8.35, −6.12	<0.001

(a) *t*-test for independent variates, comparison of total sitting and standing means between intervention and control group. Abbreviations: CI: confidence interval, SD: standard deviation, N: number, T: testing period.

**Table 4 ijerph-19-06727-t004:** Sitting and standing times in intervention and control groups clustered by sex, BMI percentiles, sports and fitness level.

		Intervention Group		Control Group		*t*-test (a)	
		*n*	M (SD)	*n*	M (SD)	95%-CI	*p*-Value
**Sex**							
Female	Time sitting	34	36.02 (3.58)	16	41.35 (2.42)	(−1.75, 1.23)	0.72
	Time standing		9.34 (3.63)		3.93 (2.40)	(−1.20, 1.80)	0.68
Male	Time sitting	52	36.05 (4.23)	22	41.09 (2.10)	(−1.71, 1.79)	0.97
	Time standing		9.34 (4.27)		4.23 (2.14)	(−1.77, 1.77)	0.99
**BMI Percentiles**							
Extreme Underweight (<3)	Time sitting	0	-	2	40.60 (0.75)	(−5.42, 2.96)	0.46
	Time standing		-		5.05 (0.26)	(−1.35, 4.64)	0.19
Underweight (3–15)	Time sitting	8	36.80 (3.33)	4	41.83 (1.96)	-	-
	Time standing		8.58 (3.40)		3.40 (1.92)	-	-
Normal weight (15–85)	Time sitting	51	35.78 (3.96)	24	40.99 (2.30)	(−2.40, 2.37)	0.99
	Time standing		9.59 (4.01)		4.28 (2.33)	(−2.48, 2.34)	0.95
Overweight (85–97)	Time sitting	11	37.12 (3.15)	5	41.01 (2.72)	(−3.89, 1.22)	0.30
	Time standing		8.25 (3.19)		4.35 (2.68)	(−1.25, 3.92)	0.31
Obesity (>97)	Time sitting	16	35.74 (4.85)	3	42.73 (1.83)	-	-
	Time standing		9.66 (4.87)		2.61 (1.85)	-	-
**Sport activities** **(outside the school setting)**	fx						
No	Time sitting	44	36.47 (3.55)	8	41.96 (2.48)	(−0.82, 2.74)	0.28
	Time standing		8.92 (3.60)		3.37 (2.55)	(−2.72, 0.86)	0.30
Yes	Time sitting	42	35.59 (4.35)	30	40.10 (2.13)	(−0.83, 2.57)	0.31
	Time standing		9.78 (4.40)		4.30 (2.13)	(−2.57, 0.87)	0.33
**Fitness Level**							
Low (<30)	Time sitting	12	39.49 (2.01)	1	44.52 (0.0)	(−1.25, 7.79)	0.15
	Time standing		5.87 (1.95)		0.81 (0.0)	(−7.79, 1.31)	0.16
Normal (30–70)	Time sitting	67	35.81 (3.72)	33	41.25 (2.18)	(−1.00, 3.64)	0.001
	Time standing		9.57 (3.76)		4.06 (2.20)	(−3,64, 1.04)	0.001
High (>70)	Time sitting	7	32.36 (4.8)	4	39.94 (1.86)	(0.42, 6.46)	0.026
	Time standing		13.11 (4.86)		5.35 (1.93)	(−6.61, −0.49)	0.024

(a) *t*-test for independent variates, comparison of total sitting and standing means between intervention and control group.

## Data Availability

Data can be made available upon request.

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
