# Peer review of "Influence of Sit-Stand Tables in Classrooms on Children’s Sedentary Behavior and Teacher’s Acceptance and Feasibility: A Mixed-Methods Study"

_ijerph, 2022, doi:10.3390/ijerph19116727_

Round 1

Reviewer 1 Report

This study discusses the impact of using sit-stand-tables on children and adolescents' sedentary behavior and teachers' acceptance and feasibility, and makes a beneficial exploration for reducing sedentary behavior. However, the following problems still exist in this study:

  1. The Move 3 accelerometer used in this study can recognize sitting and standing posture. It is necessary to introduce the principle and setting parameters of this device in detail so that other researchers can repeat it.
  2. Tables 3 and 4 of this study are not readable and need to be re integrated. For example, in Table 2, the third row is marked with mean ± SD, but the data under t-test is not mean ± SD. Besides, what is "KI"? The same is true for table 4. In Table 4, it is difficult for me to see which variables are performing t-test. The explanation of SD abbreviation appeared twice.
  3. I disagree with L401-405 in the discussion section. I think there is a correlation between the observed phenomenon and the outcome. This study cannot conclude that regular use of sit-stand-tables will help improve physical fitness, and it may be that children who are physically fit are more active themselves.
  4. Simplify the conclusion and highlight the key points.

Reviewer 2 Report

Thank you so much for inviting me to review this manuscript. Authors tried to (1) analyse sitting and standing times in classrooms with and without sit-stand-tables, (2) explore which children in classrooms with sit-stand-tables are more likely to change between sitting and standing times during lessons with a special focus on sex, BMI, active sports and fitness level, and (3) explore intervention-group teachers´ feasibility and acceptance of the intervention. Congratulations to the authors for their effort. I have only some comments:

  • Please, include the aim of the study in the abstract.
  • How was the fitness measured and categorized? This information should be included. 
  • What are active sports? Are there inactive sports? 
  • "Extreme overweight" term could be misleading.
  • What criteria were used to measure BMI? I recommend using the International Obesity Task Force criteria (as well as their categories). 

    Best regards,

Reviewer 3 Report

This is the report of a mixed-methods study on the effects of sit-and-stand desks on sitting-standing time during classes in primary and secondary school students, and teachers' perceptions of this, including the learning environment. It is a long report nevertheless fairly easy to read and understand (despite some typos and some questionable phrase construction, later highlighted) except for Tables 3 and 4 which are very dense, and I believe have some errors. Data provided in table–Appendix A would be great in a figure. I have some more prominent doubts regarding the study design and analysis (quantitative data) and less noticeable worries, more related to the quality of the reporting which I detail later.

More prominent uncertainties/recommendations:

  • How long was this study ongoing? One year? With 3 points of assessment (T0, T1 and T2)? And on each time point students wore the accelerometer for 5 days. Is that it? How long from T0 to T1 to T2? And the comparisons are just between-groups in T2? The design has impact on statistical analysis selection (quantitative data). Sample size in tables and in figure 2 (quantitative analysis, T2) are very different. At this point, I’m very confused.
  • How was the nested component integrated in sample size calculation, group allocation and/or statistical analysis?
  • Authors report that sample size calculation and sensitivity analysis was performed with G-power (lines 184–185), but don’t report any results on this.
  • Furthermore, you use simple independent t-test to assess the association of sedentary behaviour regarding sex, BMI, active sports and fitness level, but multivariate analysis may be more robust.
  • I usually don’t recommend any amendment regarding descriptives concerning time units, except when they are an important outcome measure, as in this study. reading means and SD of 8.79 (just an arbitrary number) is just too exhausting to convert mentally for minutes and seconds on a constant basis. Please consider presenting descriptives in minutes and the decimals in seconds.

Specific comments/suggestions:

Abstract

Line 20: minus sign is not correct (before 30.9 min)

Keywords

You have too many key words. Select the ones that are synonyms of those that already appear in the title. That may increase the chance of your article appear when searching in databases.

Introduction

Lines 37–38: Last part of the sentence is a little bit confusing. Do you mean inactivity times outside the school?

Line 40: Please provide a couple of examples of poorer health outcomes

Line 65: “increasing physical activity” or reduce sedentary time?

Materials and Methods

Line 108: What about children with mobility limitations, e.g., those using wheelchair for moving around?

Lines 138, 170, etc.: I prefer “data” rather than “part”.

Line 145: I prefer “Physical performance measures” over “Exercises”.

Lines 150–151: Please inform (briefly) about the questions or provide the questionnaire as appendix/supplementary material.

Line 197: I don’t like “for preparing the independent t-test analyses”. Please reword.

Results

Table 1: Please substitute “Gender” by “Sex”. Also, your total N is in the same line as the “Sex”. Please amend.

Tables 3 and 4: There is too many “mean (SD)” within the tables hindering readability. Please revise. You may state in the caption that values are presented as means and SD unless otherwise state. Also, typo on KI (German I believe) and the column of p-values look strange and too many “p-values” presented. And perhaps p-values are best suited as the last column (near the 95%CI for the mean differences).

Line 261: SD for CG is missing.

Line 313: I believe it may be “stood” instead of “stud”.

Line 351: I believe may be “larger” instead of “lager”. (I prefer a good Trappist though :p)

Discussion

Differences in sitting/standing time between intervention and control were rather expected. What is relevant for the discussion, and is absent, is how important were the time students spent more active/less sedentary in terms of potential health gains. How important or how potentially important are the ~31 minutes per day, in average, to improve health. The arguments they present regarding the obesity are not strongly convincing. Standing may requires 1.5 to 2 MET, hence just 1.75 to 3.5 mLO2/kg/min than rest or sitting (1 MET). Assuming that for each liter of O2 consumed per minute we expend 5 Kcal, 31 minutes represents, saying for a 50 Kg child, an increase in energy expenditure up to 5 Kcal (50 Kg*0.00175 LO2*31 min = 2,75 Kcal to 50 Kg*0.0035 LO2*31 min = 5,43 Kcal) which seems very little (compared to what authors are claiming in lines 409–415 regarding increased caloric expenditure)

Line 446: I don’t understand this adversity “…not fulfilled fear of disruptions in the beginning”. Maybe the “not” is confusing me…

Lines 450–451: Not sure if this final sentence fits well in this paragraph. Typo “lager”?

Conclusion

Conclusions should incorporate mostly your findings. As so, I think the lines 508–510 and 513 are best suited in the Discussion section, not in Conclusion.

Reviewer 4 Report

  1. How were measured the parameters described in figure 3 (page 12)?. It is important since the teacher´s perspective is included in the results and discussion of the work
  2. A high percentage of children (84%) included in the control group had sports activities after school, the above could impact on sitting pattern or response in this group?

Round 2

Reviewer 2 Report

Authors have resolved all of my comments. To me, the manuscript can be accepted. Only a minor comment. Please, indicate only "obesity" and no "obese". Children with obesity is more respectful than obese children. "Obese" can be pejorative. 

Congratulations for your work. 

Best wishes, 

Author Response

Response to Reviewer 2

Point 1: Authors have resolved all of my comments. To me, the manuscript can be accepted. Only a minor comment. Please, indicate only "obesity" and no "obese". Children with obesity is more respectful than obese children. "Obese" can be pejorative. 

Answer: Thank you for this comment, we changed the wording within the manuscript from "obese" to "obesity".

Thank you for your revisions and wishes!

Reviewer 3 Report

The quality of the reporting in this version has significantly improved, particularly Tables 3 and 4. I accept the amendments authors have performed in the text considering my previous comments/suggestions.

Best wishes.

Author Response

Response to Reviewer 3

Point 1: The quality of the reporting in this version has significantly improved, particularly Tables 3 and 4. I accept the amendments authors have performed in the text considering my previous comments/suggestions.

Best wishes.

Answer: Thank you for your feedback, your revisions and your wishes!